# Prevalence of COVID-19 Associated Mucormycosis in a German Tertiary Care Hospital

**DOI:** 10.3390/jof8030307

**Published:** 2022-03-17

**Authors:** Ulrike Scharmann, Frank Herbstreit, Nina Kristin Steckel, Jutta Dedy, Jan Buer, Peter-Michael Rath, Hedda Luise Verhasselt

**Affiliations:** 1Institute of Medical Microbiology, University Hospital Essen, University of Duisburg-Essen, 45122 Essen, Germany; ulrike.scharmann@uk-essen.de (U.S.); jan.buer@uk-essen.de (J.B.); peter-michael.rath@uk-essen.de (P.-M.R.); 2Department of Anesthesiology and Intensive Care Medicine, University Hospital Essen, 45122 Essen, Germany; frank.herbstreit@uk-essen.de; 3West German Cancer Center, Department of Bone Marrow Transplantation, University Hospital Essen, 45122 Essen, Germany; nina-kristin.steckel@uk-essen.de; 4Pharmacy, University Hospital Essen, 45122 Essen, Germany; jutta.dedy@uk-essen.de

**Keywords:** SARS-CoV-2, COVID-19, fungal infection, mucormycosis, Mucorales qPCR

## Abstract

Due to Coronavirus disease (COVID-19) a new group of patients at risk emerged with COVID-19-associated mucormycosis (CAM). Systematic studies, evaluating the prevalence of CAM are missing. To assess CAM prevalence in a tertiary care hospital in Germany, we applied direct microscopy, fungal culture and quantitative realtime in-house PCR targeting Mucorales-specific fragments of 18S and 28S rRNA on respiratory specimens of 100 critically ill COVID-19 patients. Overall, one Mucorales-PCR positive bronchoalevolar lavage was found whereas direct microscopy and fungal culture were negative in all cases. We conclude that a routine screening for CAM in Germany is not indicated.

## 1. Introduction

Mucormycosis are infections caused by fungi of the order Mucorales, predominantly due to *Rhizopus* spp., *Mucor* spp., and *Lichtheimia* spp., *Rhizomucor* spp. among others [1]. Cases present as rhino-cerebral, pulmonary, disseminated or cutaneous mucormycosis and most common underlying diseases or incidents are diabetic patients, patients with immunosuppressive conditions such as neutropenia after organ or stem cell transplant and immunocompetent patients after skin trauma [1]. Due to the pandemic of Coronavirus disease (COVID-19) caused by the severe acute respiratory syndrome coronavirus type 2 (SARS-CoV-2), a new group of patients at risk emerged in which secondary fungal infections are described and further increase the mortality rate: COVID-19–associated pulmonary aspergillosis (CAPA) and COVID-19–associated mucormycosis (CAM). In CAM pathogenesis, damage of lung endothelium and tissue, phagocytic cell dysfunction due to hyperglycaemia and acidosis in uncontrolled Type 2 diabetes mellitus and immune cell suppression after corticosteroid use among others play a role [2].

Before the SARS-CoV-2 pandemic, the incidence of mucormycosis was low in Europe, e.g., 0.02 cases per 100,000 population as estimated for Germany [3] compared to nearly 70 times higher than global data (estimated to be at 14 cases per 100,000 population in India) [4,5]. Due to the COVID-19 pandemic, the prevalence of mucormycosis increased 2.1 times in India [6]. In August 2021, Muthu et al. evaluated 233 reported cases from India and 42 of the rest of the world [7], of which rhino-orbital and rhino-orbito-cerebral mucormycosis were the most common infested locations (89% in India and 64% globally). Pulmonary CAM was assigned with 7.3% in India and 21.4% in the rest of the world. They found a mortality of 36.5% in India and 61.9% globally [7]. According to a meta-analysis, the pooled global prevalence of CAM accounted seven cases per 1000 patients which was 50 times higher than the highest recorded background of mucormycosis [8].

Several recommendations exist regarding clinical management and diagnosis of CAM [2,9] which are similar to non-COVID-19 patients. As mucormycosis is rapidly progressive, controlling underlying predisposing factors, surgical debridement and initiation of antifungal therapy with amphotericin is urgently necessary.

To investigate the CAM prevalence in critically ill patients we designed this study.

## 2. Materials and Methods

Respiratory specimens from 100 COVID-19 patients treated on ICUs of the University Hospital Essen from July 2020 until May 2021 were included in this study. Specimens were sent for microbiological pneumonia workup (culture and molecular detection) to the laboratory and the required minimal sample volume was 10 mL. All specimens were divided into two parts to perform microscopic examination and mycological culture as well as PCR assays. After arrival at the laboratory, respiratory samples were stored up to 48 h at 2–8 °C until DNA extraction. For DNA isolation from liquid respiratory specimen (bronchoalveolar lavage fluid (BAL); tracheal secretion (TS) and bronchial secretion (BS)) the Maxwell 16 LEV Total RNA Tissue Kit XAS1220 (Promega) and the Maxwell 16 instrument (both Promega Corp., Madison, WI, USA) were used. DNA was kept frozen at −20 °C until further analysis. The commercial multiplex real-time PCR MucorGenius (PathoNostics, Maastricht, The Netherlands) was performed in batch in June 2021 according to the manufacturer’s instructions using a Rotor-Gene Q (Qiagen, Hilden, Germany) and the Rotor-Gene Q software 2.3.5.1. M13 phage was used as internal control and the positive control was used from the kit provided by the manufacturer. All controls were within the acceptable range. Direct microscopy of BALs was conducted and specimen were plated on Malt Extract agar (Oxoid, Wesel, Germany) and incubated at 36 °C for two days following eight days at 25 °C. Plates were checked for fungal growth daily. Clinical details were analyzed retrospectively.

## 3. Results

The patients were predominantly male (71.0%) with a mean age of 58 years (Table 1). 

All patients were treated on an intensive care unit, 96% received mechanical ventilation and 50% extracorporeal membrane oxygenation. Analyzed specimens were mostly BAL (92%) followed by TS (6%) and BS (2%). For 27% of patients, Type 2 diabetes was an underlying disease and 59% were treated with steroids (dexamethasone) as recommended to treat severe COVID-19. For 15 patients, treatment with steroids and Type 2 diabetes coincided. Patients received no *Mucor*-specific antimycotic therapy (e.g., amphotericin B). 

Overall, no growth of Mucorales was observed in the entire cohort and direct microscopy revealed no fungal hyphae. However, one positive Mucorales-PCR (1.0%) with a cycle threshold (ct) value of 28.8 was found. The positive Mucorales-PCR belonged to a specimen of an 82-year old male with severe acute respiratory deficiency syndrome and three-vessel coronary artery disease as underlying disease. Initially, he presented with dyspnea (saturation of peripheral oxygen = 81% and a breathing rate of 35 per minute) in the referral hospital and piperacillin/tazobactam (4.5 g q8h i. v. as a prolonged infusion over two hours) was started. Due to respiratory and hemodynamic deterioration, he was transferred to the University Hospital Essen and received non-invasive ventilation therapy in addition to dexamethasone (6 mg OD i. v.) and remdesivir (100 mg OD i. v.). In the clinical course, he was intubated and antimicrobial therapy was expanded by caspofungin (70 mg loading dose i. v. followed by 70 mg OD) and piperacillin/tazobactam was changed to meropenem (2 g q8h i. v.) after detection of Candida glabrata and Klebsiella pneumoniae in blood culture. Chest computed tomography indicated bipulmonal ground-glass opacities with crazy paving and consolidations in the right apical superior lobe segment (CO-RADS Category 6). The total period of hospitalization was 26 days during which six independent respiratory specimens (BAL: four times and TS: two times) were analyzed by cultural approach always without fungal growth. The Mucorales-PCR positive BAL was obtained five days after admission to the University Hospital. The patient died after a second episode of bacteremia with Enterobacter cloacae in septic shock with multi-organ dysfunction. Due to the retrospective nature of the study, the positive Mucorales-PCR result was obtained after the patient’s death. 

## 4. Discussion

To the best of our knowledge, this is the first systematic approach to analyze the prevalence of COVID-associated mucormycosis in critically ill patients in Germany. Most notably, BALs as specimens with high diagnostic impact were assessed with two independent diagnostic approaches (culture and quantitative PCR). 

One of 100 investigated respiratory specimens from 100 patients showed detection of Mucorales-DNA, while fungal culture remained negative. According to the manufacturer, the observed ct value of 28.8 from the patient’s sample is positive. The applied PCR has a sensitivity of 90% and a specificity of 98% regarding pulmonary materials [10]. In general, the PCR we used simultaneously targets the 18S rDNA region of *Mucor* spp., *Rhizopus* spp., *Rhizomucor* spp. and *Cunninghamella* spp., without discriminating between genera and thus detects the most common Mucorales species [1]. 

In the global guideline for diagnosing mucormycosis, molecular methods are only moderately recommended because of less standardization whereas microscopy and culture are strongly recommended [1]. However, obtaining a lung biopsy as the golden standard for mycological examination is challenging in critically ill patients with COVID-19. Furthermore, half of cultures are false negative [11,12]. The less invasive sampling, a sensitivity of 90% of commercially available PCR, and the lower turnaround time of PCR from respiratory specimens offer clear advantages over fungal culture approach in diagnosing CAM especially in a pandemic situation. In a large CAM cohort from France, Mucorales-PCR was found positive in 15 (88%) patients, mostly in serum (*n* = 14) and BAL (*n* = 7) and PCR was the only successful detection method for 4 patients [13].

In India, localized forms of mucormycosis predominate not only in non-COVID-19 patients with hematological diseases (72% [14]) second to lung involvement (58%) but also in patients with COVID-19 (up to 90% with rhino-orbito-cerebral mucormycosis and <10% pulmonary disease [6,7]). In Germany, pulmonary mucormycosis were more frequent than rhino-orbito-cerebral mucormycosis [15]. Seidel et al. investigated the prevalence of CAM in Germany. They found 13 cases from six tertiary hospitals in Germany between January 2020 and June 2021, and they depicted a higher prevalence of CAM in only two centers (2 out of 100,000 and 8 out of 100,000) [15]. Up to now, systematic studies for the prevalence of CAM from Europe are missing. In our cohort, there was no clinical suspicion of localized mucormycosis, which would have implied further workup. 

The treatment with steroids and the prevalence of diabetes as well as malignancies found in our cohort reflects the group at risk for infection with Mucorales. However, the positive Mucorales-PCR was found in a patient under steroid treatment without diabetes or cancer but he received antifungal treatment with caspofungin for candidemia. In patients with leukemia and stem cell transplantation recipients receiving *Aspergillus*-active drugs (echinocandins or voriconazole) mucormycosis has been increasingly observed [16] due to intrinsic resistance of Mucorales. 

Consistent with others, our findings underline the importance of clinical/radiological/microbiological assessment in patients with predispositions for COVID-associated mucormycosis but due to the low prevalence, a routine screening seems not to be indicated currently. However, multicenter studies are desirable for substantiation of findings. 

Our study has some limitations: from the patient with positive Mucorales-PCR we examined six respiratory specimens by culture but only one by PCR. Since histology or PCR obtained by a sterile aspiration or biopsy from a pulmonary site was not obtained, performance of molecular diagnostic from multiple samples would have been helpful to verify the fungal infection. 

## Figures and Tables

**Table 1 jof-08-00307-t001:** Demographics and underlying conditions of patients with COVID-19 examined for Mucorales.

	No. Patients (%)
Total	100 (100)
Type of specimen from respiratory tract: BAL, TS, BS	92 (92), 6 (6.0), 2 (2.0)
Filamentous fungi-specific culture Ɨ performed/positive for Mucorales	100 (100.0)/0
Mucorales PCR # performed/positive	100 (100)/1 (1.0)
Hospitalized	100 (100)
Treatment on intensive care unit	100 (100)
Mean age (years)	58
Male	71 (71)
Invasive mechanical ventilation	96 (96)
ECMO	50 (50)
Use of Steroids	59 (59)
Mortality	59 (59)
Underlying conditions	
Cardiovascular disease	35 (35)
Diabetes	27 (27)
Pulmonary disease	11 (11)
Cancer	4 (4.0)
Other (Crohn’s disease)	1 (1.0)
None	48 (48)
One underlying condition	31 (31)
Two underlying conditions	17 (17)
>two underlying conditions	4 (4.0)

BAL, bronchoalveolar lavage fluid; TS, tracheal secretion; BS, bronchial secretion; ECMO, extracorporeal membrane oxygenation; Ɨ Malt Extract agar (Oxoid, Wesel, Germany); # MucorGenius (PathoNostics, Maastricht, The Netherlands).

## Data Availability

The datasets generated during the current study are available from the corresponding author on request.

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
