# Peer review of "Prevalence of COVID-19 Associated Mucormycosis in a German Tertiary Care Hospital"

_jof, 2022, doi:10.3390/jof8030307_

Round 1

Reviewer 1 Report

• Study presents a good brief background of the CAM epidemiology, however, I suggest authors include the recent systematic reviews in this area in the introduction and discussion section, where it fits, such as, "Hussain S et al. J Fungi (Basel). 2021;7(11):985. 10.3390/jof7110985"

• Please reframe the below sentence in line 145, currently it is hard to comprehend.

"leading to verify the fungal infection as a proof via autopsy wasn’t conducted."

• Can the author discuss how they arrived at the sample size and were there any exclusions of patient samples once they were included.

• Do the author see scope for future similar studies in this area to confirm the finding that, "CAM prevalence is low in Germany." This can be one of the additional recommendations from this paper.

Author Response

Reviewer 1

  1. Study presents a good brief background of the CAM epidemiology, however, I suggest authors include the recent systematic reviews in this area in the introduction and discussion section, where it fits, such as, "Hussain S et al. J Fungi (Basel). 2021;7(11):985. 10.3390/jof7110985".

We now included the systematic review from Hussain et al.

  1. Please reframe the below sentence in line 145, currently it is hard to comprehend. "leading to verify the fungal infection as a proof via autopsy wasn’t conducted."

We now reframed the sentence to „Because histology or PCR obtained by a sterile aspiration or biopsy from a pulmonary site was not obtained, performance of molecular diagnostic from multiple samples, would have been helpful to verify the fungal infection.“

3 Can the author discuss how they arrived at the sample size and were there any exclusions of patient samples once they were included.

We now included a sentence about sample collection and exclusions in the Materials and Methods section.

  1. Do the author see scope for future similar studies in this area to confirm the finding that, "CAM prevalence is low in Germany." This can be one of the additional recommendations from this paper.

We are thankful for this point. We now added a sentence in the discussion that multicenter studies are desirable to underline our findings.

Reviewer 2 Report

Manuscript ID: jof-1609409

Title: Prevalence of COVID-19 associated mucormycosis in a German tertiary care hospital

General comments:

The article reports the established finding of low prevalence of mucormycosis in Germany as studies identified before. However, the article needs to be improved, because many of the basic information needs to be added.

Specific comments:

  1. Authors should italicize the Genus and species name throughout the manuscript.
  2. In introduction, authors may write a statement about estimated incidence of mucormycosis in Germany was 0.02 cases per 100,000 population (Runhke et al. 2015) and comparing with estimated incidence of higher case reporting countries such as Iran and India before the COVID-19 pandemic. Then authors can mention the incidence of mucormycosis has increased during the pandemic. To substantiate authors findings.
  3. In introduction, to investigate the CAM prevalence in critically ill patients we designed this retrospective study. Do authors mentioned prospective study as retrospective, because authors have done a study based on molecular diagnosis of mucormycosis for a particular period. Do authors collected the samples prospectively and then clinical details retrospectively. Please clarify.
  4. If authors have done molecular diagnosis from preserved samples retrospectively, this makes the diagnosis results tough to interpret. As preserved specimens always have less percentage positivity in mucormycosis diagnosis, with this it is tough to estimate the exact incidence and prevalence. Preserved samples show only 50% positivity in histopathological proven mucormycosis cases.
  5. Materials and methods: do authors use any internal controls (human genes), to check the extracted DNA from the sample was of good quality and showed amplification of the internal control for comparison with 18S and 28S RNA. If so please mention it in methods and results. Without internal control the study result is invalid.
  6. Please add information for all the above-mentioned points, if authors doesn’t have these information, then study results have lot of imitations which should be addressed in the discussion section. In discussion, please add the sensitivity and specificity of Mucorales PCR positivity from stored specimens, internal control results are must.

Author Response

Reviewer 2

  1. Authors should italicize the Genus and species name throughout the manuscript.

We thank the reviewer for this point. We now italicize genus and species names throughout the manuscript.

  1. In introduction, authors may write a statement about estimated incidence of mucormycosis in Germany was 0.02 cases per 100,000 population (Ruhnke et al. 2015) and comparing with estimated incidence of higher case reporting countries such as Iran and India before the COVID-19 pandemic. Then authors can mention the incidence of mucormycosis has increased during the pandemic. To substantiate authors findings.

We thank the reviewer for this comment. We now rephrased the incidence part in the introduction and included the Ruhnke et al. reference. 

  1. In introduction, to investigate the CAM prevalence in critically ill patients we designed this retrospective study. Do authors mentioned prospective study as retrospective, because authors have done a study based on molecular diagnosis of mucormycosis for a particular period. Do authors collected the samples prospectively and then clinical details retrospectively. Please clarify.

We now described the study type and progress more detailed.

  1. If authors have done molecular diagnosis from preserved samples retrospectively, this makes the diagnosis results tough to interpret. As preserved specimens always have less percentage positivity in mucormycosis diagnosis, with this it is tough to estimate the exact incidence and prevalence. Preserved samples show only 50% positivity in histopathological proven mucormycosis cases.

We thank the reviewer for this point. We now stated in the manuscript that we extracted DNA from respiratory samples within 48 hours after arrival in the institute. The MucorGenius qPCR was performed in batch in the first week of June 2021.

  1. Materials and methods: do authors use any internal controls (human genes), to check the extracted DNA from the sample was of good quality and showed amplification of the internal control for comparison with 18S and 28S RNA. If so, please mention it in methods and results. Without internal control the study result is invalid.

We are thankful for this comment. After rechecking the raw data files, it was apparent that the PCR we used to acquire the data presented in the initially submitted manuscript was not the in-house PCR as mistakenly stated but the commercially available MucorGenius PCR from PathoNostics. We apologise for this mistake which was not done on purpose. In routine diagnostics, we partially used in-house and MucorGenius PCR in parallel and due to an error in internal communication this wrong information got entry into the manuscript. Of course, we provide the original PCR data on request for transparency and verification. The use of internal and positive control was added to the manuscript in the Material and Method section.

  1. Please add information for all the above-mentioned points, if authors don’t have these information, then study results have lot of imitations which should be addressed in the discussion section. In discussion, please add the sensitivity and specificity of Mucorales PCR positivity from stored specimens, internal control results are must.

We now included information for all the above-mentioned points especially regarding the internal control. No explicit data can be found for the sensitivity and specificity of Mucorales PCR positivity from stored specimens, only from stored DNA eluates (Guegan et al.: Evaluation of MucorGenius® mucorales PCR assay for the diagnosis of pulmonary mucormycosis. Journal of Infection 81 (2020) 311–317. https://doi.org/10.1016/j.jinf.2020.05.051). We now added this information.

Round 2

Reviewer 2 Report

Major comment:

Authors have addressed the previously raised comments. The kit they have used was compared on proven, probable or possible mucormycosis cases. However in this study all the classes are possible cases except one. Authors should have included proven cases of mucormycosis samples (with FFPE  Specimens if fresh cases were not available) to validate the results in their laboratory settings, before proceeding to the possible cases. Authors should have compared with the in house PCR method. It would have helped the scientific community to use the kit in their laboratory settings with confidence. The major drawback of this study as authors themselves mentioned the kit does not help in species identification.

Minor comment:

  1. Line 41. estimated to be at 0.02 to 9.5 cases per 100,000 population, change it to “estimated to be at 14 cases per 100,000 population in India”.
  2.  In results authors should mention the range of DNA they obtained after DNA extraction from kit ( eg. as low as 100 µg/µl to 1000 µg/µl) and how much quantity they used in PCR process.

Author Response

Major comment:

Authors have addressed the previously raised comments. The kit they have used was compared on proven, probable or possible mucormycosis cases. However in this study all the classes are possible cases except one. Authors should have included proven cases of mucormycosis samples (with FFPE  Specimens if fresh cases were not available) to validate the results in their laboratory settings, before proceeding to the possible cases. Authors should have compared with the in house PCR method. It would have helped the scientific community to use the kit in their laboratory settings with confidence. The major drawback of this study as authors themselves mentioned the kit does not help in species identification.

We thank the reviewer for the comment. All of our investigated cases were possible cases, which is due to the prospective design of our study. In the past, we used an in-house PCR in addition to the commercially available PCR. Because of very rare proven cases, we were not able to validate the in-house PCR and therefore handled the data very carefully. This is the reason why we now use the data produced by the PCR from Pathonostics only. This PCR is CE certified and thus validated for use in diagnostic settings. In their review on molecular diagnosis of mucormycosis, Lackner et al. assessed a sensitivity of 90% and a specificity of 98% for the PCR from Pathonostics for pulmonary samples (Lackner N, Posch W, Lass-Flörl C. Microbiological and Molecular Diagnosis of Mucormycosis: From Old to New. Microorganisms. 2021 Jul 16;9(7):1518. doi: 10.3390/microorganisms9071518. PMID: 34361953; PMCID: PMC8304313.)

Minor comment:

  1. Line 41. estimated to be at 0.02 to 9.5 cases per 100,000 population, change it to “estimated to be at 14 cases per 100,000 population in India”.

We now changed the sentence to “estimated to be at 14 cases per 100,000 population in India”.

  1. In results authors should mention the range of DNA they obtained after DNA extraction from kit (eg. as low as 100 µg/µl to 1000 µg/µl) and how much quantity they used in PCR process.

We thank the reviewer for this comment. We followed the instructions of the Pathonostics PCR kit. The Maxwell purification kit was used to extract genomic DNA from the samples. The extracted DNA concentration is not routinely measured due to the qualitative approach of this PCR. We now measured the DNA concentration of the positive patient sample from our study which was 77.6 ng/µl. From two negative patient samples from our study we obtained a DNA concentration of 23 and 10.2 ng/µl.